# Access to HIV Antiretroviral Therapy among People Living with HIV in Melbourne during the COVID-19 Pandemic

**DOI:** 10.3390/ijerph182312765

**Published:** 2021-12-03

**Authors:** Dooyeon Lee, Eric P. F. Chow, Ivette Aguirre, Christopher K. Fairley, Jason J. Ong

**Affiliations:** 1Melbourne Medical School, the University of Melbourne, Parkville, VIC 3010, Australia; dooyeonl@student.unimelb.edu.au; 2Melbourne Sexual Health Centre, Alfred Health, Melbourne, VIC 3053, Australia; eric.chow@monash.edu (E.P.F.C.); I.Aguirre@alfred.org.au (I.A.); CFairley@mshc.org.au (C.K.F.); 3Central Clinical School, Monash University, Melbourne, VIC 3004, Australia; 4Centre for Epidemiology and Biostatistics, Melbourne School of Population and Global Health, The University of Melbourne, Melbourne, VIC 3010, Australia; 5Faculty of Tropical and Infectious Diseases, London School of Hygiene and Tropical Medicine, London WC1E 7HT, UK

**Keywords:** HIV, antiretroviral therapy, viral load, COVID-19, pandemics, Australia

## Abstract

The social measures taken to control the COVID-19 pandemic can potentially disrupt the management of HIV. The objective of this study was to examine the impact of the Australian COVID-19 lockdown restrictions on access to antiretroviral therapy (ART) for people living with HIV in Melbourne. Using data from the Melbourne Sexual Health Centre (MSHC), we assessed the changes in rates of ART postal delivery, controlled viral load, and ART dispensing from 2018 to 2020. The percentage of ART delivered by postage from the MSHC pharmacy was calculated weekly. The percentage of people living with HIV with a controlled viral load (≤200 copies/mL) was calculated monthly. We calculated a yearly Medication Possession Ratio (MPR). The average percentage of HIV ART dispensed through postage for the years 2018, 2019, and 2020 was 3.7% (371/10,023), 3.6% (380/10,685), and 14% (1478/10,765), respectively (P_trend_ < 0.0001). Of the 3115 people living with HIV, the average MPR for 2018, 2019, and 2020 was 1.05, 1.06, and 1.14, respectively (P_trend_ = 0.28). The average percentage of people with an HIV viral load of <200 copies/mL for the years 2018, 2019, and 2020 was 97.6% (2271/2327), 98.0% (2390/2438), and 99.2% (2048/2064), respectively (P_trend_ < 0.0001). This study found that the proportion of controlled viral load and access to ART of people living with HIV in Melbourne was largely unaffected by the COVID-19 lockdown restrictions. This suggests that some of the services provided by the MSHC during the pandemic, such as HIV ART postal delivery, may assist long-term HIV management.

## 1. Introduction

Severe acute respiratory syndrome coronavirus 2 (SARS-CoV-2) is the viral agent responsible for coronavirus disease 2019 (COVID-19) and was first reported in Australia in January 2020. In March 2020, the Australian government implemented lockdown restrictions in response to the pandemic which involved the closure of Australia’s international borders, the implementation of social distancing rules, and the closure of non-essential services. The cases of COVID-19 began to decline from April 2020, and most lockdown restrictions were gradually eased from May 2020. However, in July 2020, there was a second wave of COVID-19 cases in Victoria, and the previous lockdown restrictions were reintroduced in Victoria with further restrictions, including a five-kilometre travel restriction, mandatory face masks, and the 8 p.m.–5 a.m. curfew. By 31 December 2020, there were 28,408 confirmed cases and 909 deaths from COVID-19 infection. Several studies have described reduced access to sexual health clinics during the pandemic; however, it is unclear whether there was an impact on people living with HIV (PLHIV), particularly in terms of access to antiretroviral therapy (ART) [1,2,3,4].

The current management for PLHIV involves high adherence to ART and regular monitoring of their HIV viral load. Strict adherence to ART is essential for maintaining an undetectable viral load, which reduces the risk of progressing to acquired immunodeficiency syndrome (AIDS) and the transmission of HIV to sexual partners [5]. The findings on the interruption of ART dispensing during the COVID-19 pandemic are conflicting. One study reported no significant difference in the number of ART collection visits in South African clinics, while another study reported a 23.1% reduction in the number of patients with ART scripts during the pandemic in Italy; however, this might be due to different COVID-19 situations and restrictions across countries [6,7]. In terms of impact on the viral load, a study conducted in Rome described no significant change in the proportion of undetectable viral load during the lockdown [8]. In Australia, a study reported that 98% of PLHIV in Victoria had self-reported access to ART; however, there are currently no published studies using a direct measure from medical clinics on the impact of the COVID-19 pandemic and lockdown restrictions on medication access and viral load control of PLHIV in high-income countries [9]. Furthermore, a systematic review demonstrated that most of the current studies show that mail-order pharmacies in the United States are associated with a higher medication adherence [10]. However, there has been limited study on the efficacy of the medication delivery in the context of PLHIV during the Australian COVID-19 pandemic.

This study aimed to examine whether the COVID-19 pandemic and the Victorian lockdown restrictions affected the management of PLHIV by using data from the Melbourne Sexual Health Centre (MSHC) before and during the COVID-19 pandemic lockdowns to compare the proportion of ART postage, access to ART, and the proportion of controlled viral load for PLHIV.

## 2. Materials and Methods

This study was conducted at the MSHC, the largest public HIV and STI clinic in Victoria, Australia. The MSHC has an onsite pharmacy that dispenses HIV ART scripts free of charge for MSHC patients and patients receiving HIV care at general practice (GP) clinics. For the purpose of this study, a script contains medications that are part of the patient’s HIV ART regimen, and this may be a single medication containing multiple active agents or a combination of multiple medications.

During the lockdown periods, MSHC provided both face-to-face consultations and telephone consultations to maintain access to the health service for PLHIV whilst also reducing their risk of potential exposure to SARS-CoV-2. Patients were given the option to collect HIV ART from their local community pharmacy (subject to a copayment of AUD 6.60–41.30 per two months of each medication) or to continue receiving free HIV ART from the MSHC pharmacy, with an option for the postal delivery of HIV ART at a fee equivalent to the cost of the postage satchel (AUD 16–26).

The study population included patients who had HIV ART scripts dispensed from the MSHC pharmacy, including patients from MSHC’s HIV clinic and patients from GP clinics. Moreover, the majority of the study population were males and stratifying our data by gender did not alter our conclusions.

A waiver of informed consent was granted for the use of routinely collected clinical data.

### 2.1. Study Variables

The primary study outcomes were the proportion of HIV ART delivered by postage, access to ART, and the proportion of controlled viral load (≤200 copies/mL) during the study period.

The total number of MSHC and GP clinic patients who were dispensed HIV ART scripts from the MSHC pharmacy between 2018 and 2020 was extracted from the pharmacy dispensing software (iPharmacy v9.1-DXC Technology, Sydney, Australia). HIV ART delivered by postage was recorded using a paper-based system at the MSHC pharmacy.

HIV ART dispensing data containing the providing clinic (MSHC or GP), dispensing date, medication name, and amount of medication dispensed for each patient were obtained for 2018–2020 from the MSHC pharmacy dispensing software (iPharmacy v9.1-DXC Technology, Sydney, Australia).

We also extracted electronic data on viral load among PLHIV who had their HIV care and management at MSHC between 2018 and 2020. These data only included patients from the MSHC HIV clinic and did not include the patients from GP clinics as their viral load tests were not performed at MSHC.

### 2.2. Statistical Analysis

We calculated the weekly number and the proportion of HIV ART delivered by postage during the study period. Segmented linear regression analysis was conducted to examine the trends in the proportion of HIV ART delivered by postage before (from 1 January 2018 to 22 March 2020) and during (from 23 March 2020 to 31 December 2020) the COVID-19 pandemic. A chi-squared trend test was used to examine the changes in the proportion of HIV medication delivered by postage between 2018 and 2020.

We used Medication Possession Ratio (MPR) as a measure of access to ART. MPR calculates the proportion of a timeframe where the patient had access to their ART. An MPR of greater than or equal to one suggests that the patient had adequate access to ART during that timeframe. MPR was calculated using the following formula: [11]
(1)$$\mathrm{MPR}=\frac{\mathrm{Number}\;\mathrm{of}\;\mathrm{days}\;\mathrm{covered}\;\mathrm{by}\;\mathrm{ART}\;\mathrm{between}\;\mathrm{the}\;\mathrm{first}\;\mathrm{and}\;\mathrm{last}\;\mathrm{dispense}\;\mathrm{date}}{\mathrm{Number}\;\mathrm{of}\;\mathrm{days}\;\mathrm{between}\;\mathrm{the}\;\mathrm{first}\;\mathrm{and}\;\mathrm{last}\;\mathrm{dispense}\;\mathrm{date}}$$

We used the following approach to clean the HIV ART dispensing data to calculate the MPR. If the same antiretroviral treatment was recorded multiple times on the same date for the same patient, these recordings were combined as a single dispensing, and the quantity was multiplied by the number of times it was dispensed. If a combination of antiretroviral treatments were dispensed on the same date for the same patient, they were considered as a single dispensing of combination ART. For example, if one month’s worth of medication Abacavir–Lamivudine, Darunavir–Cobicistat, and Dolutegravir were dispensed on the same date, this was considered one month’s worth of combination therapy, not three months’ worth of medications (Appendix A). Only the individuals with HIV ART scripts dispensed from the cleaned database in 2018, 2019, and 2020 were included in the study.

If the patient only had a single ART script dispensed in a year, the MPR was calculated using the first dispense date in that year and the first dispense date in the following year. This means some MPR calculations may include data from the following year, but this would show whether or not a single script for one year was sufficient until the patient’s next script in the following year. Additionally, if the patient only had a single ART script dispensed in 2020 that had enough medication to cover the rest of the year, the MPR was recorded as one. If the patient only had a single ART script dispensed in 2020 that did not have enough for the rest of the year, the MPR was calculated by considering the end date as 31 December 2020. We calculated the mean MPR each year to assess the impact of the pandemic and lockdown restrictions on treatment access for PLHIV. Then, we separated the dispensing data into PLHIV receiving their ART from MSHC and those receiving their ART from the GP clinic and compared the calculated MPR. Furthermore, we conducted a sensitivity analysis using an alternative approach for MPR calculation by dividing the total number of days covered by every ART script in a year by 365.

To measure the changes in viral load over time, we restricted the analysis by including individuals who had viral loads measured in the last three years (2018 to 2020). The viral loads were categorised into controlled (≤200 copies/mL) and uncontrolled (>200 copies/mL), and the proportion of individuals with controlled viral loads was calculated for each month. A chi-squared trend test was used to examine the changes in the annual proportion of controlled viral loads between 2018 and 2020. We calculated the mean proportion of controlled viral load and used an interrupted time series analysis to observe changes in the proportion of controlled viral load before and during the COVID-19 pandemic. We conducted sensitivity analyses for the MPR and viral load by only including those who had measurements in both 2019 and 2020 to observe any changes to the interpretation of the results. All statistical analyses were conducted using STATA (version 14) (StataCorp, College Station, TX, USA). This study was approved by the Alfred Hospital Ethics Committee, Melbourne, Australia (773/20).

## 3. Results

Between 2018 and 2020, a total of 31,473 HIV ART scripts were dispensed from the MSHC pharmacy for 4551 PLHIV (9916 scripts for 3734 PLHIV in 2018, 10,727 scripts for 3830 PLHIV in 2019, and 10,830 for 3877 PLHIV in 2020). In total, 18,162 scripts were dispensed for patients from GP clinics, and 13,311 scripts were dispensed for the MSHC patients. Of those, 2229 (7.1%) HIV ART were delivered by postage, and the annual proportion of HIV ART delivered by postage increased from 3.7% (371/10,023) in 2018 to 3.6% (380/10,685) in 2019 to 14% (1478/10,765) in 2020 (P_trend_ < 0.0001). Figure 1 shows the trend that the proportion of HIV ART delivered by postage did not change significantly before and during the COVID-19 pandemic; however, there was a 140% immediate increase in the proportion of HIV ART delivered by postage in the week starting 23 March 2020 (i.e., the first week after the implementation of the lockdown restrictions). Refer to Appendix A for the proportion of HIV ART delivered by postage superimposed with the weekly COVID-19 cases during 2020.

Of the total of 31,473 HIV ART scripts dispensed among 4551 patients, 26,666 scripts were dispensed for 3115 PLHIV who had ART scripts dispensed in all three years—2018, 2019, and 2020 (8569 in 2018, 9134 in 2019, and 8963 in 2020) (Table 1). In 2020, 211 PLHIV only had one script, including 48 PLHIV with enough ART and 163 PLHIV with not enough ART in 2020. The average MPR for the years 2018, 2019, and 2020 was 1.05 (95% CI: 0.99 to 1.11), 1.06 (95% CI: 0.98 to 1.13), and 1.14 (95% CI: 0.99 to 1.28), respectively (P_trend_ = 0.28). Of the 26,666 ART scripts dispensed, 16,026 (60%) were for patients from GP clinics and 10,640 (40%) were for MSHC patients. The average MPR for patients from GP clinics for 2018, 2019, and 2020 was 1.03 (95% CI: 0.92 to 1.14), 1.06 (95% CI: 0.92 to 1.19), and 1.08 (95% CI: 0.94 to 1.22), respectively. The average MPR for patients from the MSHC for 2018, 2019, and 2020 was 1.07 (95% CI: 1.05 to 1.09), 1.06 (95% CI: 1.02 to 1.09), and 1.21 (95% CI: 0.94 to 1.48), respectively. Furthermore, the average MPR calculated using the alternative method, the total number of days covered by every ART script in a year divided by 365, for 2018, 2019, and 2020 were 0.92 (95% CI: 0.91 to 0.93), 0.94 (95% CI: 0.94 to 0.95), and 0.92 (95% CI: 0.91 to 0.93), respectively (Appendix A).

There was a total of 6829 viral load measurements between 2018 and 2020 for 1123 PLHIV from MSHC with viral load measurements in all three years—2018, 2019, and 2020. There was a reduction in the total number of viral load measurements from 2327 (2018) and 2438 (2019) to 2064 (2020). The proportion of controlled viral load (≤200 copies/mL) in 2018, 2019, and 2020 were 97.6% (2271/2327), 98.0% (2390/2438), and 99.2% (2048/2064), respectively (P_trend_ < 0.0001). Figure 2 shows the trends in the proportion of controlled viral load that remained stable before and during the COVID-19 pandemic.

The sensitivity analyses are reported in Appendix A. The interpretation of the result was not affected by examining those who had measurements in both 2019 and 2020.

## 4. Discussions

This study is the first to investigate the impact of the COVID-19 pandemic and the lockdown restrictions on both ART access and viral load measurements of PLHIV. This study demonstrated that during the year 2020, there was a significant increase in the proportion of HIV ART delivered through postage and no declines in the proportion of people with controlled viral load and medication access compared to the previous years. Furthermore, the average MPR remained above one between 2018 and 2020. These findings suggest that the management of the people living with HIV at the MSHC was largely unaffected by the COVID-19 pandemic, which concurs with the current understanding of the impact of the pandemic on HIV management in Melbourne [9].

We found an increased preference for ART postage during the COVID-19 pandemic: the proportion of ART postal delivery significantly increased from 3.7% in 2018 and 3.6% in 2019 to 14% in 2020. This postal service option was important during the COVID-19 pandemic to minimise the risk of exposure and address the patients’ concerns for contracting the coronavirus while visiting the clinic.

There was no difference in the MPR during the COVID-19 pandemic, and the mean MPR was above one in 2018, 2019, and 2020 for GP clinics and the MSHC. Despite some concerns about the potential shortage and interruption of HIV ART during the pandemic, the results show that ART supplies were adequate in Melbourne during the pandemic [2,3,12]. Furthermore, this suggests that the HIV ART postal service provided by the MSHC was a beneficial service for the PLHIV in Melbourne. HIV ART postal service should be actively implemented, especially in areas where remote ART delivery was not readily available for people living with HIV during the COVID-19 pandemic [13].

The proportion of PLHIV with a controlled viral load did not decline during the COVID-19 pandemic. Several additional services were provided by the MSHC during the COVID-19 pandemic to monitor and manage PLHIV, such as telehealth and HIV ART postal delivery service. Furthermore, there was a reduction in the total number of viral load measurements conducted in 2020. Possible reasons for this are postponing HIV blood tests during the pandemic for those with a long-term history of viral load suppression, which meant that they did not require the six-monthly viral load measurements, or an increased number of blood tests conducted outside of the MSHC.

This study shows that ART management of people living with HIV in Melbourne continued uninterrupted during the COVID-19 pandemic. Our findings are consistent with another Australian study involving an online survey for 153 PLHIV in Victoria, which reported 98% of PLHIV having self-reported access to ART during the pandemic [9]. Moreover, this study describes no change in the proportion of controlled viral load in Melbourne, which is in alignment with the findings in Rome, Italy [8].

The strength of this study is that it is the first to quantitatively assess the impact of the pandemic on the HIV ART management of PLHIV through both MPR and viral load level.

There are some limitations to this study. First, we did not know whether PLHIV also collected their HIV ART medication from other pharmacies outside the MSHC. If this occurred, it would have underestimated the MPR calculation. Second, this study was conducted based on data from the MSHC and therefore may not be generalisable to all services in Australia, in settings with larger numbers of COVID-19 cases or more severe lockdown restrictions. Third, there is a potential for selection bias given the reduction in the absolute number of viral load tests performed in 2020. Although the MSHC ensured that the six-monthly viral load testing continued for those that needed it, the individuals without a viral load measurement in 2020 may be those with an uncontrolled viral load level. Moreover, we did not exclude the viral loads of the newly diagnosed PLHIV as this information was not explicitly available. This may have biased the proportion of controlled viral load. However, the number of newly diagnosed HIV cases at the MSHC between 2018 and 2020 was small (i.e., approximately 60 cases per year), which account for 5.3% of all patients included in the viral load analysis and is unlikely to affect the interpretation of the results [14,15]. Fourth, the MPR calculation for 2020 may have been overestimated for those receiving ART beyond 2020 as the end date was assumed to be 31/12/2020. Fifth, by only including PLHIV with dispensing records in 2018, 2019, and 2020 for the MPR calculation, the study may be omitting those that did not pick up scripts due to the pandemic. However, based on the stable number of PLHIV receiving ART scripts (3734 in 2018, 3830 in 2019, and 3877 in 2020) and the MSHC report showing a stable number of individuals visiting the HIV clinic between 2018 and 2020, this is less likely to affect the interpretation of the results [16]. Sixth, although the proportion of ART delivered by postage increased and the MPR and viral load were not negatively affected, the direct causal relationship between the study variables was not evaluated in this study. Lastly, this study could not investigate the psychosocial factors that may have influenced MPR and viral load measurements.

## 5. Conclusions

In conclusion, the management of people living with HIV at the MSHC was largely unaffected during the COVID-19 pandemic. The ART postal service increased in popularity during the COVID-19 pandemic and might have contributed to the high percentage of people with a controlled viral load even when the strictest restrictions related to COVID-19 were being enforced.

## Figures and Tables

**Figure 1 ijerph-18-12765-f001:**
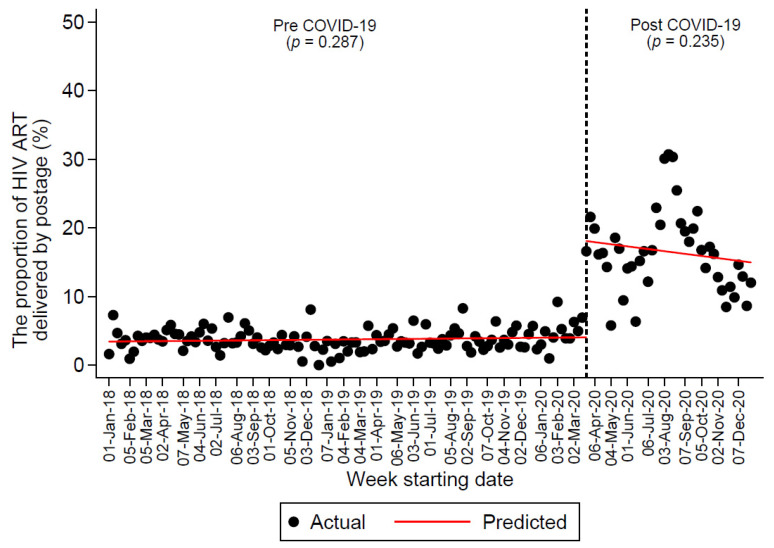
Segmented linear regression analysis of the proportion of HIV ART delivered by postage before and during the COVID-19 pandemic (indicated by the dotted line).

**Figure 2 ijerph-18-12765-f002:**
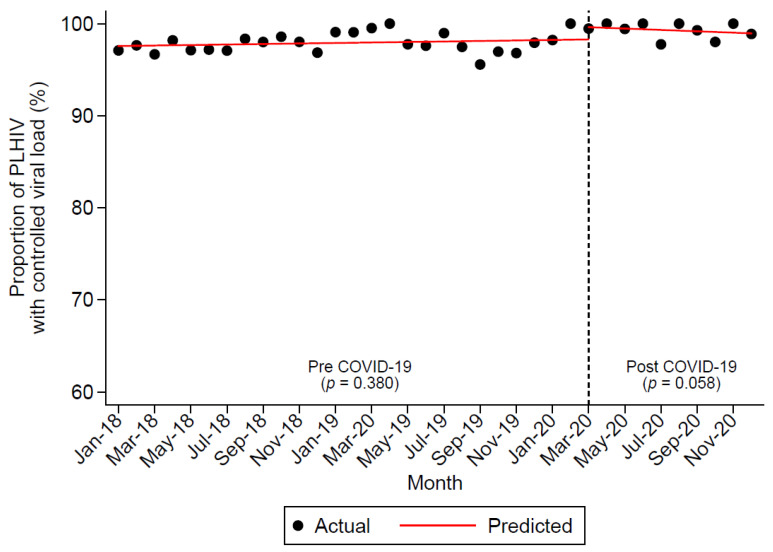
Segmented linear regression analysis of the percentage of undetectable viral load before and during the COVID-19 pandemic (indicated by the dotted line).

**Table 1 ijerph-18-12765-t001:** Medication Possession Ratios.

	Years	Mean (SD)	95% Confidence Interval	Median	IQR
**GP and MSHC Patients**	2018	1.05 (1.81)	0.99 to 1.11	0.99	0.91–1.07
	2019	1.06 (2.23)	0.98 to 1.13	0.99	0.91–1.07
	2020	1.14 (4.11)	0.99 to 1.28	0.99	0.90–1.07
**GP patients**	2018	1.03 (2.40)	0.92 to 1.14	0.98	0.90–1.05
	2019	1.06 (2.92)	0.92 to 1.19	0.98	0.89–1.05
	2020	1.08 (3.05)	0.94 to 1.22	0.98	0.89–1.06
**MSHC patients**	2018	1.07 (0.44)	1.05 to 1.09	1.02	0.93–1.12
	2019	1.06 (0.72)	1.02 to 1.09	1.02	0.93–1.09
	2020	1.21 (5.12)	0.94 to 1.48	1.01	0.92–1.08

GP = general practice; IQR = interquartile range; MSHC = Melbourne Sexual Health Centre; SD = standard deviation.

## Data Availability

All relevant data are presented in the manuscript. Further details can be obtained from the corresponding author on reasonable request.

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
