# Peer review of "Access to HIV Antiretroviral Therapy among People Living with HIV in Melbourne during the COVID-19 Pandemic"

_ijerph, 2021, doi:10.3390/ijerph182312765_

Round 1

Reviewer 1 Report

Comments to Author

This manuscript by Lee D et.al was conducted to investigate if COVID-19 lockdown restrictions affects the viral load and access to HIV antiretroviral medication among people living with HIV in Melbourne. The manuscript is well performed and well written.

Minor comments

  1. Line no151-156, please represent it as graphical format (yearly 2018, 2019 and 2020) to show its the trends as written in the text.
  2. From line 240-256, please provide it as a dedicated section as limitation of this study.

Reviewer 2 Report

Lee and coworkers in their report investigated the accessibility of HIV antiretroviral therapy during the COVID-19 pandemic. The area of investigation was Melbourne. The study is important especially in context to COVID 19 pandemic however, some questions need to be addressed:

  1. There are two important aspects of the study:
  2. Accessibility of HIV-1 therapy to the affected patients
  3. COIVD 19 era

The authors thoroughly talked about point ‘a’ however, there is no data provided about point ‘b’. What was the severity rate of COVID19 at that period in which HIV antiretroviral data was assessed? This data must be provided as it will truly reflect the relationship between the COVID era and HIV management.

The authors should provide the rate of HIV therapy received by males and females and should discuss the data in the COVID 19 context. The study is important and is well within the scope of the study. I suggest the resubmission of the article addressing the above points.

Reviewer 3 Report

This manuscript describes the shift in dispensing of HIV ART at the Melbourne Sexual Health Centre during the first wave of the CoVID-19 pandemic to include about a 10% increase to those receiving their medication by postal delivery. No differences in viral load or ART access were detected in patient data collected before and during the pandemic lockdown restrictions. These results are inline with similar studies with different study sites that have previously been reported.

The research design is appropriate and very well described. The results are clearly presented and support the conclusions. The main weakness of this paper is that it lacks originality or interesting results.

It may improve the introduction to add discussion on the efficacy of postal pharmacy dispensing in the context of the existing literature on the topic.

The main weakness of the study is the lack of originality or interesting results. While the increase in postal delivery of medication during lockdown is somewhat novel, it is not unexpected. It cannot be discerned with this study design if postal delivery had effects on ART compliance or was simply convenient for study participants. Similar studies have reported similar outcomes for ART access during CoVID-19 lockdowns.

While the manuscript is nicely prepared and comprehensive, given the shortcoming of novel findings, it may be more impactful to reduce the content to the form of a brief report.

Reviewer 4 Report

None.

Author Response

No comments or suggestions were provided by Reviewer 4.

Round 2

Reviewer 2 Report

The authors have tried to improve the manuscript. This manuscript could have a great impact if authors could dig some COVID19 infection data at the time point at which HIV-1 data was collected. This could be merged with Figure 1. In case if it is not possible please include this one of the limitations of the study. The study is important as it is addressing HIV-1 in COVID 19. The manuscript is well within the scope of the journal and may be accepted for publication after the minor change (suggested above). I congratulate the team on their good work.
